# Impact of Combined Exposure to Glyphosate and Diquat on Microbial Community Structure and Diversity in Lateritic Paddy Soil

Xiaoyu He [1,2], Chunyuan Wu [2,3,4,5], Huadong Tan [2,3,4,5,*], Xiao Deng [2,3,4,5,*] and Yi Li [2,3,4,5]

1 National Key Laboratory of Green Pesticide, Key Laboratory of Green Pesticide and Agricultural Bioengineering, Ministry of Education, Center for R&D of Fine Chemicals of Guizhou University, Guiyang 550025, China; gs.xiaoyuhe20@gzu.edu.cn
2 Environment and Plant Protection Institute, Chinese Academy of Tropical Agricultural Sciences, Haikou 571101, China; wuchunyuangz@126.com (C.W.); wish.0310@163.com (Y.L.)
3 Danzhou Scientific Observing and Experimental Station of Agro-Environment, Ministry of Agriculture and Rural Affairs, Danzhou 571737, China
4 National Agricultural Experimental Station for Agricultural Environment, Danzhou 571737, China
5 Hainan Engineering Research Center for Non-Point Source and Heavy Metal Pollution Control, Danzhou 571737, China
* Correspondence: tanhuadong1991@163.com (H.T.); dx0928@foxmail.com (X.D.)

**Abstract:** Microbial communities play crucial roles in the biogeochemical cycling of many important soil elements. Pesticides are known to affect non-targeted soil microorganisms. Glyphosate (GP) and diquat (DQ), two commonly used non-selective herbicides, often co-exist in lateritic paddy soil rich in iron/aluminum oxides. However, there is limited information on their co-impact on microbial community structure and diversity in this type of soil. Here, the short-term effects of combined exposure to GP and DQ on microbial diversity and community structure shifts were investigated in lateritic paddy soil from a tropical agricultural region (Hainan, China). The combined utilization dosages of two herbicides were set in three concentrations: low concentration (1 fold of the recommended dosage), medium concentration (10 fold of the recommended dosage) and high concentration (100 fold of the recommended dosage). The structure and diversity of microbial communities were determined via 16S rRNA and ITS gene high-throughput sequencing. The results revealed that Actinobacteria and Proteobacteria were the most sensitive microbial phyla to the combined exposure of GP and DQ in lateritic paddy soil. The combined exposure to GP and DQ increased the abundance of Actinobacteria but significantly inhibited that of Proteobacteria, especially at low and medium concentrations. Compared with CK, mixed herbicide (GP + DQ) had no adverse effects on the richness of bacteria and fungi communities as well as on the diversity of bacteria communities, but it significantly decreased the diversity of fungi communities at high concentrations within 28 days. However, the effects of combined exposure to GP and DQ on soil microbial richness and diversity were not significantly different from those of separate exposure of the two herbicides. In conclusion, the combined application of GP and DQ had no more adverse effects on soil microorganisms. Therefore, these two herbicides can be used reasonably in actual agricultural production.

**Keywords:** herbicides; mixed toxicity; microbial diversity; microbial community structure

## 1. Introduction

Herbicides are the most widely used class of pesticides in global agriculture [1], with up to 1.2 million tons in 2018 [2]. Among herbicides, glyphosate (GP) and diquat (DQ), as two non-selective herbicides, both rank in the top five most commonly used herbicides in the global market; in particular, GP has the highest use in the global market with 0.86 million tons used in 2016 [3]. At present, GP was recorded to cause heavy pollution in different

environmental media [4–6], e.g., water (21.2–32.5 μg/L) [7–9], soil (1200–1502 μg/kg) [10–12] and sediment (1149 μg/kg) [13], especially in tropical soil (690–40,000 μg/kg) [12,14–17]. So far, however, there have been few studies on the appearance, persistence, and distribution of DQ in the environment; the residue of DQ in sediments is obviously larger than that in soil [18]. Meanwhile, lateritic paddy soil is rich in iron and aluminum, due to its strong binding with clay particles and organic matter, and DQ is easily retained in the lateritic paddy soil; even the adsorption of GP in soils increased with iron and aluminum oxides content [19–21]. In practice, two or more mixed herbicides (e.g., GP and DQ) are often applied in varied agricultural scenarios, especially in the tropics [6,16,22]. Undoubtedly, the GP and DQ will inevitably co-occur in lateritic paddy soil. However, there is limited information on the co-occurrence of GP and DQ on microbial diversity and community structure shifts in lateritic paddy soil.

Microbial communities in the soil environment play an important role in the biogeochemistry cycle of many important soil elements [23,24]. It was found that herbicide residues changed the soil microflora and had a potentially long-term effect on the functions of agricultural soils [25–28]. Most studies have shown that GP has no or transient stress effect on the soil microbial community [29–34], but some studies have shown significant adverse effects [35–37]. For instance, GP can lead to a decrease in the abundance of *Pseudomonas fluorescens*, Mn-transforming bacteria, and indoleacetic acid-producing bacteria in soybean rhizosphere soil [38]. The abundance of certain Bacteroidetes, Chloroflexi, Cyanobacteria, Planctomycetes, and alpha-Proteobacteria members are highly negatively correlated with GP concentrations [39]. There is also some other literature on the specific functional or ecological microbial groups in different GP exposure cycles, including the effects of GP on Acidobacteria, ammonia-oxidizing bacteria, Mycorrhiza, etc. [40–42]. Due to the strong binding property of DQ with clay particles, it is reasonable to assume that higher levels of DQ occurred in lateritic paddy soil, which might bring more pressure on the structure and function of the microbe. Nevertheless, there are few studies on the effects of combined exposure of GP with other herbicides on microbial populations [43], where the combined toxicology of GP with other chemicals on microorganisms mainly focuses on it with heavy metals or polyethylene microplastics [44–49]. GP contains coordination groups such as carboxyl, amino, and phosphate groups, and has a strong complexing ability to heavy metal cations and organic cations [46]. Similar to the charge characteristics of metal ions, DQ has the ability to strongly adsorb on the surface with negative charges and also has a strong oxidation–reduction cycling ability that can be reduced to generate free radicals [19]. These similar characteristics coupled with the observed difference (i.e., strong oxidation–reduction cycling ability) might partly contribute to the combined toxic effects of GP and DQ differing from that of GP and heavy metal.

Are soil microbial communities and community structures significantly affected by the combined use of herbicides in lateritic paddy soil in tropical agricultural areas? We hypothesized that mixed herbicides would have significant adverse effects on the overall community structure and diversity of soil microorganisms; in addition, the effects of mixed herbicides on soil microbiota were significantly different from those of single herbicides, and mixed herbicides can reduce or increase the abundance of certain microbial communities in the soil. In the context, the short-term exposure (28 days) test was conducted, wherein the effects of single GP, DQ, and their mixture on the diversity and community structure of soil bacteria and fungi were investigated in lateritic paddy soil from a tropical agricultural region (Hainan, China) based on 16S rRNA and Internal Transcribed Spacer (ITS) high-throughput sequencing technology. We hope the results of the study will provide a scientific basis for the rational composite application of herbicides in lateritic paddy soil in agricultural areas.

## 2. Materials and Methods

### 2.1. Experiment Design and Sample Collection

Lateritic paddy soils (0–15 cm) were collected from a village (19°55′ N, 110°25′ E) in the tropical agricultural region of the Nandu River Basin in Hainan Province, China. The residues of GP and DQ were not found in the soil (the limits of detection were <0.03 mg kg$^{-1}$) based on the analytical methods by Delhomme et al. [50] and Pizzutti et al. [51]. The texture of the soil is a typical sandy loam identified using a method from USDA [52]. In order to recover soil microorganisms, the soil samples were sieved through a sieve (2 mm). Then, the deionized water was added to keep the soil at 50% of the maximum water holding capacity, lasting 2 weeks for pre-incubation. Analytical grade DQ and GP (active ingredient >98.5%) were purchased from Beijing Tanmo Quality Inspection Technology Co., Ltd. (Beijing, China). The recommended field dose of GP was 0.6 mg kg$^{-1}$, and meanwhile, the recommended dosage of DQ in the field was 0.4 mg kg$^{-1}$ [53]. In the experiment, 10 treatments were set up based on the recommended field dose, containing a blank only with deionized water and the 9 different experimental treatments of GP, DQ, and GP + DQ at different concentrations (Table 1). The initial weight of soil in each treatment was 50 g. During the experiment, deionized water was added regularly during the incubation process to compensate for evaporated water. Overall, 3 g samples were collected from each treatment on days 1, 7, 14 and 28 after application; then, the soil samples were stored at −80 °C and taken out for testing within 3 days. All experiments were conducted in triplicates.

**Table 1.** Trial treatment setting. The low concentration (L) was 1 time the recommended dosage, the medium concentration (M) was 10 times the recommended dosage, and the high concentration (H) was 100 times the recommended dosage. The recommended dosages were set as 0.6 and 0.4 mg kg$^{-1}$ for glyphosate (GP) and diquat (DQ), respectively.

| Treatment | Concentration (mg kg$^{-1}$) | | |
|---|---|---|---|
| | Low (l) | Middle (m) | High (h) |
| Blank (CK) | | 0 | |
| Glyphosate (G) | 0.6 (Gl) | 6 (Gm) | 60 (Gh) |
| Diquat (D) | 0.4 (Dl) | 4 (Dm) | 40 (Dh) |
| Glyphosate + diquat (GD) | 0.6 + 0.4 (GDl) | 6 + 4 (GDm) | 60 + 40 (GDh) |

### 2.2. DNA Extraction and Database Construction

Microbial DNA was extracted using the HiPure Soil DNA Kits (or HiPure Stool DNA Kits) (Magen, Guangzhou, China) according to the manufacturer's protocols. The 16S rDNA target region of the ribosomal RNA gene was amplified via PCR (95 °C for 5 min, followed by 30 cycles at 95 °C for 1 min, 60 °C for 1 min, and 72 °C for 1 min and a final extension at 72 °C for 7 min) using primers listed in Table S1. PCR reactions were performed in triplicate using the 50 μL mixture containing 10 μL of 5 × Q5@ Reaction Buffer, 10 μL of 5 × Q5@ High GC Enhancer, 1.5 μL of 2.5 mM dNTPs, 1.5 μL of each primer (10 μM), 0.2 μL of Q5@ High-Fidelity DNA Polymerase, and 50 ng of template DNA. Related PCR reagents were from New England Biolabs, Ipswich, MA, USA.

For bacteria, after genomic DNA was extracted from the soil samples, the 16S rDNA V3 + V4 region was amplified with barcode-specific primers. The primer sequence was: 341F: CCTACGGGNGGCWGCAG; 806R: GGACTACHVGGGTATCTAAT.

For fungi, the amplified region is the ITS2 region of ITS. The primer sequences were as follows: ITS3: GATGAAGAACGYAGYRAA; ITS4: TCCTCCGCTTATTGATATGC. The purified amplified products were connected to sequencing linkers to construct sequencing libraries and sequenced using Illumina.

### 2.3. Bioinformatic Analysis of Microbial 16S rRNA and ITS

After the raw data named "Reads" were obtained via sequencing, the low-quality data in Reads were filtered first; then, the two-terminal Reads were spliced into Tag, and the Tag was filtered again. The data obtained were called Clean Tag. Next, based on Clean Tag, Usearch software was used to cluster, remove the chimeric Tag detected during the clustering process, and obtain the representative sequences and abundance of OTU. The representative OTU sequences were classified into organisms via a naive Bayesian model using an RDP classifier [54] (version 2.2) based on SILVA database [55] (version 132) or UNITE database [56] (version 8.0) or ITS2 database [57] (version update 2015), with the confidence threshold value of 0.8. Based on the sequence and abundance data of OTU, species annotation, species composition analysis, indicator species analysis, α diversity analysis and β diversity analysis were carried out. In α diversity analysis, the Chao1 index and the Shannon's evenness index were calculated in QIIME [58] (version 1.9.1). The alpha index comparison between groups was calculated using the Welch's *t*-test and Wilcoxon rank test in the R project Vegan package [59] (version 2.5.3). The alpha index comparison among groups was computed using the Tukey's HSD test and Kruskal–Wallis H test in the R project Vegan package [60] (version 2.5.3). If there was effective grouping, the differences between groups were compared and tested statistically.

### 2.4. Data Analysis and Statistical Analysis

Gene abundance data were analyzed using One-way Analysis of Variance (ANOVA) by using the SPSS Statistical Package (version 19.0, IBM, Armonk, NY, USA). The Duncan's multi-range test and Spearman's rank statistical analysis were used to calculate the correlation between samples, bacteria and fungi and BDGs/PDGs. All data are mean ± standard error (SE) of three replicates. The data were considered to be significant when $p < 0.05$.

## 3. Results

### 3.1. Effects of Combined Exposure of GP and DQ on the Composition and Diversity of Soil Bacterial Community

3.1.1. Bacterial Community Composition

In general, it was observed that GP decreased the relative abundance of Actinobacteria; nevertheless, DQ had no significant effect on Actinobacteria. Particularly, the combined pollution of GP and DQ increased the relative abundance of Actinobacteria, as shown in Figure 1a. Specifically, a low concentration of mixed herbicides caused the relative abundance of Actinobacteria to increase by 4.91%, 6.78% and 4.25% on the 7th, 14th and 28th day, respectively; a medium concentration of mixed herbicides caused the relative abundance of Actinobacteria to increase by 3.98% and 5.21% on the 14th and 28th day, respectively; and a high concentration of mixed herbicides caused the relative abundance of Actinobacteria to increase by 4.38% and 2.37% on the 7th and 14th day, respectively. Notably, the mixed herbicides enhanced the inhibitory effect of DQ on Proteobacteria, although the GP increased the abundance of Proteobacteria and the DQ decreased the abundance of Proteobacteria, as shown in Figure 1b. Specifically, a low concentration of mixed herbicides caused the relative abundance of Proteobacteria to decrease by 3.58% and 5.70% on the 7th and 14th day, respectively; a medium concentration of mixed herbicides caused the relative abundance of Proteobacteria to decrease by 4.93% and 2.76% on the 14th and 28th day, respectively; and a high concentration of mixed herbicides caused the relative abundance of Proteobacteria to decrease by 4.97%, 2.49% and 0.92% on the 7th, 14th and 28th day, respectively. Taken together, the results indicated that combined pollution of GP and DQ could increase the relative abundance of Actinobacteria and decrease the relative abundance of Proteobacteria compared with a single herbicide.

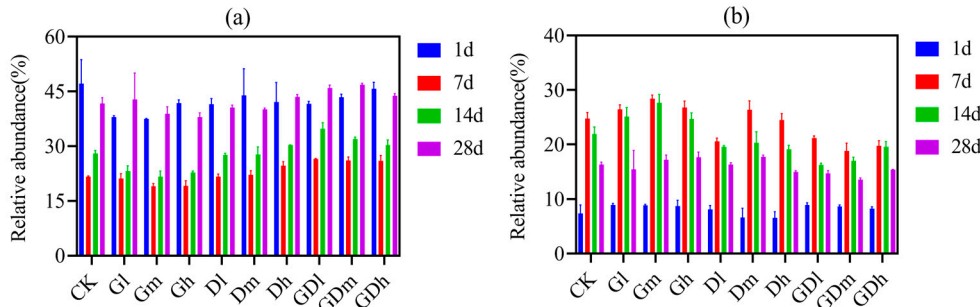

**Figure 1.** The abundance distribution of Actinobacteria (**a**) and Proteobacteria (**b**). CK—control, Gl—low concentration of glyphosate, Gm—medium concentration of glyphosate, Gh—high concentration of glyphosate, Dl—low concentration of diquat, Dm—medium concentration of diquat, Dh—high concentration of diquat, GDl—low concentration of mixed herbicides, GDm—medium concentration of mixed herbicides, and GDh—high concentration of mixed herbicides.

The stacking chart of the distribution of bacterial community composition at the genus level is shown in Figure 2a–d. Overall, it could be observed that GP increased the relative abundance of *Sphingomonas* while DQ decreased the abundance of *Sphingomonas*, whereas the inhibition of mixed herbicides on *Sphingomonas* was stronger than that of DQ, but the inhibitory effect weakened over time. Specifically, a low concentration of mixed herbicides caused the relative abundance of *Sphingomonas* to decrease by 4.36%, 3.39% and 1.40% on the 7th, 14th and 28th day, respectively; a medium concentration of mixed herbicides caused the relative abundance of *Sphingomonas* to decrease by 5.13%, 3.11% and 1.92% on the 7th, 14th and 28th day, respectively; while the high concentration of mixed herbicides caused the relative abundance of *Sphingomonas* to decrease by 4.30%, 1.41% and 0.64% on the 7th, 14th and 28th day, respectively. Remarkably, the mixed herbicides increased the relative abundance of *Streptomyces*, although an inhibiting effect on the *Streptomyces* by GP and DQ was observed. Specifically, a low concentration of mixed herbicides caused the relative abundance of *Streptomyces* to increase by 0.69%, 0.86% and 0.81% on the 7th, 14th and 28th day, respectively; while the medium concentration of mixed herbicides caused the relative abundance of *Streptomyces* to increase by 0.85%, 0.69% and 1.29% on the 7th, 14th and 28th day, respectively. In addition, it was observed that GP increased the relative abundance of *Phenylobacterium*; nevertheless, DQ had little effect on *Phenylobacterium*. Particularly, the mixed herbicides decreased the relative abundance of *Phenylobacterium* to some extent. On the 7th day, a low concentration of mixed herbicides caused the abundance of *Phenylobacterium* to decrease by 0.59%. However, the medium concentration of mixed herbicide reduced the *Phenylobacterium* abundance by 0.74% on day 7. The results showed that the mixed herbicides inhibited the growth of *Sphingomonas* and *Phenylobacterium*, and promoted the growth of *Streptomyces* at low and high concentrations as compared with a single herbicide.

### 3.1.2. Alpha Diversity of the Bacterial Community

The results of bacterial richness and diversity analysis are shown in Table 2. Generally, the Chao1 index was used to evaluate microbial richness, while the Shannon index was used to evaluate microbial diversity. The larger the Chao1 index is, the higher the microbial richness will be. In addition, the larger the Shannon index is, the higher the microbial diversity will be. On day 1 and day 28, it was observed that the medium concentration of GP increased bacterial richness (increased by 413.413/146.912) while the same concentration of DQ inhibited bacterial richness (reduced by 195.023/48.648). Particularly, the medium concentration of mixed herbicides promoted bacterial richness (increased by 300.247/63.717). It is noteworthy that medium concentrations of GP and DQ promoted bacterial diversity on days 7 and 14, but by day 28, the impact of single and mixed herbicides on soil bacterial diversity was not significantly different. In general, the effects of

mixed herbicides on soil bacterial richness and diversity were not significantly different from those of a single herbicide.

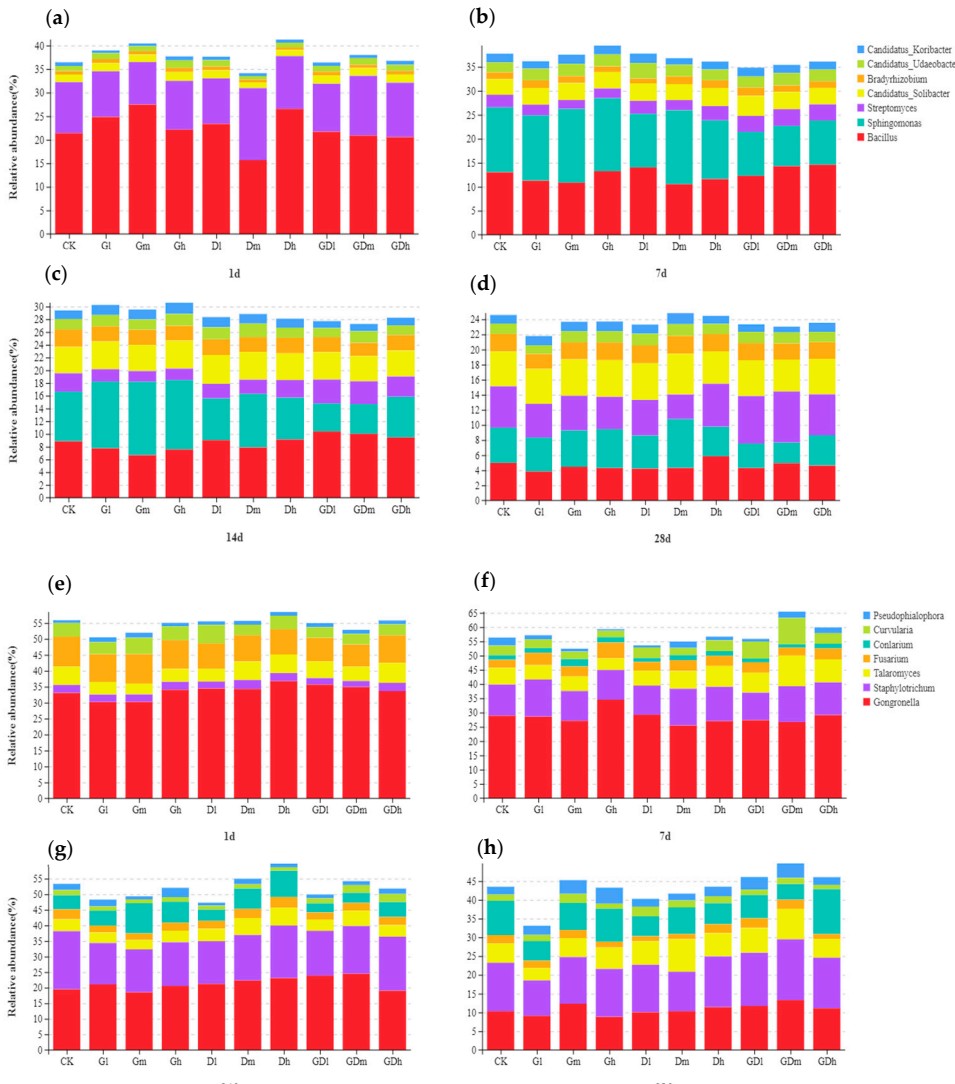

**Figure 2.** Community composition of bacteria (**a**–**d**) and fungi (**e**–**h**) at the genus level from different exposure time (1, 7, 14 and 28 d).

### 3.1.3. Beta Diversity of the Bacterial Community

The principal coordinate analyses based on the Bray–Curtis distance for the bacterial communities at the phylum level and genus level are shown in Figures 3a,b and S1. The results showed that the difference in community structure between the control and the composite herbicide was less than that between the control and the single herbicide (Figure S1). In addition, on day 1 (Figure 3a) and day 14 (Figure 3b), the difference in community structure between the mixed herbicide treatment and the control treatment was less than that between the single herbicide treatment and the control treatment ($R^2 = 0.723$, $p = 0.001$; $R^2 = 0.405$, $p = 0.044$). From the above analysis, it can be concluded that the effect of mixed herbicides on the soil bacterial community structure was less than that of single herbicides at low concentrations, although there was no significant difference in the effects between mixed herbicides and single herbicides on the soil bacterial community structure at medium or high concentrations.

**Table 2.** Alpha diversity of bacterial communities under different treatments. Different letters indicate significant differences at the $p < 0.05$ level between different treatments at the same time. Mean values ($n = 3$) ± S.E.

| Index | Treatment | Days after Application | | | |
|---|---|---|---|---|---|
| | | 1 | 7 | 14 | 28 |
| Shannon | CK | 5.918 ± 0.311 ab | 7.264 ± 0.027 bc | 7.556 ± 0.023 cd | 7.315 ± 0.093 abc |
| | Gl | 6.390 ± 0.071 a | 7.458 ± 0.009 a | 7.650 ± 0.017 abc | 7.254 ± 0.320 bc |
| | Gm | 6.249 ± 0.063 a | 7.405 ± 0.032 ab | 7.736 ± 0.022 a | 7.497 ± 0.090 ab |
| | Gh | 6.266 ± 0.161 a | 7.145 ± 0.051 c | 7.671 ± 0.018 ab | 7.453 ± 0.067 abc |
| | Dl | 6.131 ± 0.140 a | 7.407 ± 0.033 ab | 7.583 ± 0.015 bcd | 7.416 ± 0.027 abc |
| | Dm | 5.382 ± 0.379 b | 7.485 ± 0.044 a | 7.669 ± 0.030 ab | 7.629 ± 0.006 a |
| | Dh | 5.770 ± 0.256 ab | 7.416 ± 0.038 ab | 7.404 ± 0.011 e | 7.152 ± 0.014 bc |
| | GDl | 6.298 ± 0.046 a | 7.381 ± 0.029 ab | 7.284 ± 0.078 f | 7.100 ± 0.034 c |
| | GDm | 6.409 ± 0.001 a | 7.414 ± 0.090 ab | 7.505 ± 0.015 d | 7.090 ± 0.023 c |
| | GDh | 6.092 ± 0.104 a | 7.370 ± 0.106 ab | 7.499 ± 0.038 de | 7.238 ± 0.039 bc |
| Chao1 | CK | 2188.649 ± 150.191 bcd | 2263.584 ± 36.558 ab | 2369.947 ± 39.031 a | 2303.254 ± 45.996 ab |
| | Gl | 2536.324 ± 75.475 ab | 2377.208 ± 3.108 a | 2352.648 ± 36.608 a | 2219.027 ± 135.281 b |
| | Gm | 2602.062 ± 60.436 a | 2338.966 ± 56.674 a | 2386.601 ± 49.563 a | 2450.166 ± 44.036 aA |
| | Gh | 2508.073 ± 91.128 abc | 2184.617 ± 30.267 b | 2417.438 ± 28.661 a | 2285.848 ± 30.396 ab |
| | Dl | 2229.911 ± 153.341 bcd | 2406.547 ± 28.774 a | 2334.005 ± 44.040 a | 2352.122 ± 40.203 ab |
| | Dm | 1993.626 ± 203.891 d | 2417.230 ± 8.741 a | 2351.966 ± 34.642 a | 2254.606 ± 47.180 b |
| | Dh | 2154.460 ± 98.706 cd | 2329.326 ± 48.987 ab | 2249.354 ± 70.180 a | 2297.709 ± 26.360 ab |
| | GDl | 2372.901 ± 44.448 abc | 2332.078 ± 103.682 ab | 2305.535 ± 69.564 a | 2309.014 ± 8.035 ab |
| | GDm | 2488.896 ± 9.119 abc | 2363.224 ± 52.350 a | 2358.002 ± 63.922 a | 2366.971 ± 23.944 ab |
| | GDh | 2250.869 ± 64.077 abcd | 2380.824 ± 14.352 a | 2332.714 ± 39.074 a | 2250.471 ± 9.346 b |

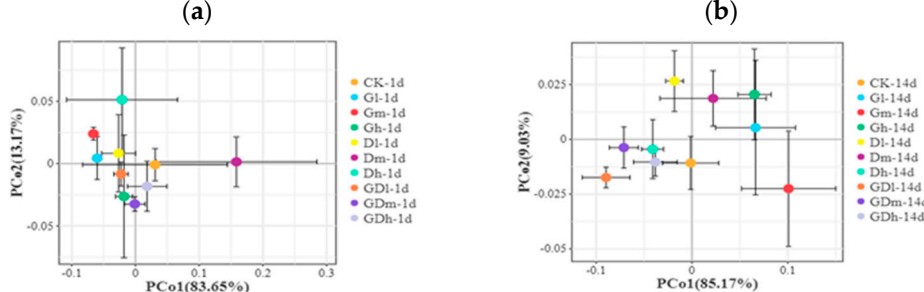

**Figure 3.** Principal coordinate analysis for the bacterial communities at the phylum level at the 1—day (**a**) and 14—day (**b**) exposure.

### 3.1.4. LEfSe Analysis of the Bacterial Community

Linear discriminant analysis (LDA ≥ 2) of the bacterial community is shown in Figures 4a and S2a–h. It can be observed that at the phylum level, Cyanobacteria were the significantly different biomarker taxa between GP treatment and other herbicide treatments on day 1 ($p \leq 0.038$). On the 14th day, the significantly different biomarker taxa between GP treatment and other herbicide treatments became Bacteroides ($p \leq 0.040$) instead of Cyanobacteria. It is important to note that Chloroflexi is sensitive to various herbicides treatments on day 14. Meanwhile, at the genus level, *Streptomyces* showed a significant difference between mixed herbicide treatments and other herbicides treatments on day 14 ($p \leq 0.034$). However, on day 28, the significantly different biomarker taxa between GP treatment and other herbicide treatments became *Rickettsia* ($p \leq 0.013$) rather than *Streptomyces*. In particular, *Streptomyces* was sensitive to mixed herbicide responses on day 14, while *Rickettsia* was sensitive to GP responses on days 14 and 28. From the above analysis, it can be concluded that the relative abundance of Cyanobacteria, Bacteroides, and *Rickettsia* is suppressed after adding DQ to the GP. In particular, compared to a single herbicide, mixed herbicides significantly increased the abundance of *Streptomyces* ($p < 0.05$).

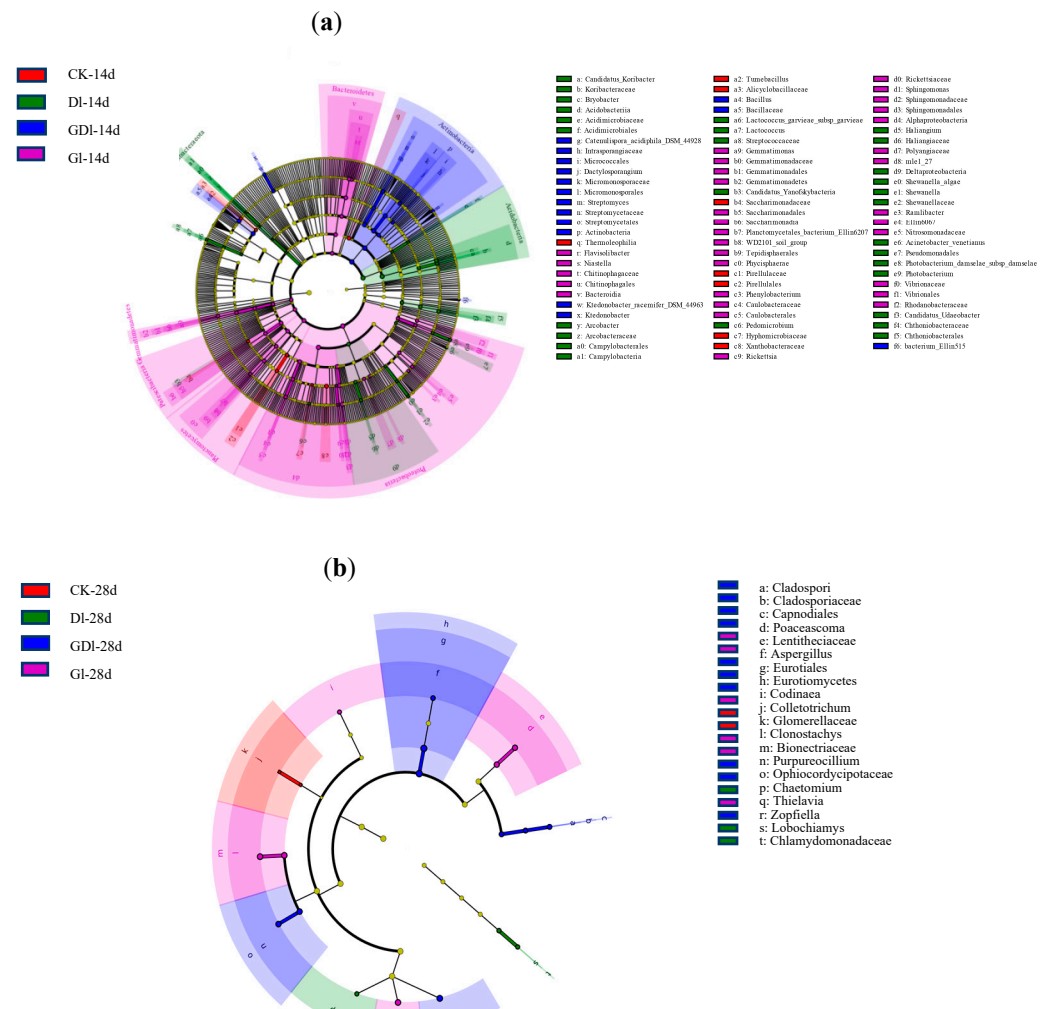

**Figure 4.** Cladograms of line discriminant analysis effect size (LEfSe) analyses based on microbial community composition in different treats (The line discriminant analysis (LDA) scores above the preset value of 2.0 were considered to be significant. From the center outward, the circles represent the different taxonomic levels of the bacteria and fungi from the phylum to the genus levels. The yellow circles denote the taxa without significant differences among the different soil layers). (**a**) bacteria, (**b**) fungi.

## 3.2. Effects of Combined Exposure of GP and DQ on the Composition and Diversity of the Soil Fungal Community

### 3.2.1. Fungal Community Composition

The stacking chart of the species distribution of fungi at the phylum level is shown in Figure 5. Notably, at the phylum level, the low concentration of mixed herbicides inhibited the abundance of Basidomycota but promoted the abundance of Ascomycota on the 7th and 28th day compared with a single herbicide. Specifically, on day 7, a low concentration of mixed herbicide reduced the abundance of Basidomycota by 3.3%, while increased the Ascomycota abundance by 4.86%; in addition, the abundance of Basidomycota decreased by 3.21% and that of Ascomycota increased by 2.33% on day 14. Taken together, the results showed that Basidomycota and Ascomycota were sensitive to exposure to mixed herbicides at certain exposure times, especially on days 7 and 14.

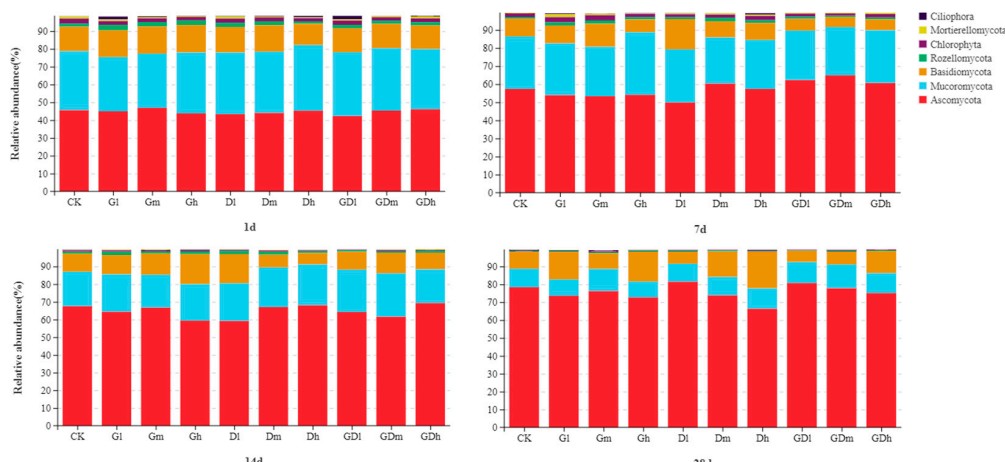

**Figure 5.** Community composition of fungi at the phylum level from different exposure times (1, 7, 14 and 28 d).

Figure 2e–h shows a stacked distribution of fungi at the genus level. In general, it was observed that GP inhibited the abundance of *Talaromyces* at low and middle concentrations on the 7th and 28th day; nevertheless the same concentrations of DQ promoted the abundance of *Talaromyces* at the same time. Notably, the addition of GP enhanced the effect of DQ on the abundance of *Talaromyces*. Specifically, a low concentration of mixed herbicides increased the abundance of *Talaromyces* by 1.16% and 1.43%, respectively, on the 7th and 28th day; while the medium concentration of mixed herbicides increased the abundance of *Talaromyces* by 4.91% and 2.95%, respectively, on the 7th and 28th day. However, on the 7th and 14th day, the mixed herbicides promoted the abundance of *Curvularia*, although the single herbicide inhibited the abundance of *Curvularia*. Specifically, the low concentration of mixed herbicides increased the abundance of *Curvularia* by 2.60% on the 7th and 14th day; the medium concentration of mixed herbicides increased the abundance of *Curvularia* by 5.85% and 0.86% on the 7th and 14th day, respectively; while the medium concentration of mixed herbicides increased the abundance of *Curvularia* by 0.30% and 0.97% on the 7th and 14th day, respectively. Particularly, on the 7th, 14th and 28th days, the mixed herbicides inhibited the abundance of *Conlarium* significantly compared with single herbicides at low and medium concentrations, and the inhibitory effect became more significant over time. Specifically, the low concentration of mixed herbicides, respectively, reduced the abundance of *Conlarium* by 0.42%, 1.32% and 5.01% on the 7th, 14th and 28th day, while the medium concentration of mixed herbicides reduced the abundance of *Conlarium* by 1.26%, 1.60% and 2.97% on the 7th, 14th and 28th day, respectively. The results showed that, compared with single herbicides, the mixed herbicides promoted the abundance of *Talaromyces* and *Curvularia* at low and medium concentrations, but inhibited the abundance of *Conlarium*.

### 3.2.2. Alpha Diversity of Fungal Community

The fungal richness and diversity data are presented in Table 3. It is observed that the diversity of fungal communities increased with time, but the richness decreased over time. On the first day, the fungal diversity of low and medium concentration GP treatments was significantly higher than that of mixed herbicide treatments at the same concentration ($p < 0.05$). Particularly, on the 14th day, low concentrations of single herbicides had little impact on fungal diversity, while low concentrations of mixed herbicides significantly inhibited fungal diversity ($p < 0.05$). The high concentration of mixed herbicides has little impact on fungal diversity; nevertheless, at the same concentration, the fungal diversity of GP treatment is significantly lower than that of the control treatment ($p < 0.05$), while the fungal diversity of DQ treatment is significantly higher than that of control treatment ($p < 0.05$). It is noteworthy that on day 28, high concentrations of mixed herbicides significantly inhibited fungal diversity ($p < 0.05$), while single herbicides at the same concentration

had no significant impact on fungal diversity. It can be observed that compared to the single herbicides, mixed herbicides inhibit the diversity of fungal communities in soil, but have no significant impacts on the richness of fungal.

**Table 3.** Alpha diversity of fungal communities under different treatments. Different letters indicate significant differences at the $p < 0.05$ level between different treatments at the same time. Mean values ($n = 3$) ± S.E.

| Index | Treatment | Days after Application | | | |
|-------|-----------|------|------|------|------|
| | | **1** | **7** | **14** | **28** |
| Shannon | CK | 4.780 ± 0.083 abc | 4.368 ± 0.046 cd | 4.614 ± 0.015 a | 4.727 ± 0.044 abc |
| | Gl | 4.882 ± 0.054 a | 4.634 ± 0.023 ab | 4.363 ± 0.029 ab | 4.783 ± 0.039 ab |
| | Gm | 4.765 ± 0.011 ab | 4.635 ± 0.086 bc | 4.259 ± 0.135 ab | 4.980 ± 0.057 a |
| | Gh | 4.544 ± 0.129 d | 4.231 ± 0.043 d | 4.388 ± 0.121 ab | 4.684 ± 0.023 cd |
| | Dl | 4.686 ± 0.072 bcd | 4.371 ± 0.020 cd | 4.259 ± 0.076 ab | 4.708 ± 0.007 bcd |
| | Dm | 4.683 ± 0.018 abc | 4.652 ± 0.044 ab | 4.516 ± 0.052 ab | 4.661 ± 0.028 bcd |
| | Dh | 4.509 ± 0.014 cd | 4.800 ± 0.053 a | 4.488 ± 0.043 ab | 4.598 ± 0.053 cd |
| | GDl | 4.688 ± 0.033 abc | 4.405 ± 0.049 cd | 4.128 ± 0.111 b | 4.867 ± 0.076 bcd |
| | GDm | 4.452 ± 0.014 d | 4.325 ± 0.028 cd | 4.376 ± 0.097 ab | 4.821 ± 0.134 abcd |
| | GDh | 4.763 ± 0.060 abc | 4.490 ± 0.069 cd | 4.369 ± 0.010 ab | 4.547 ± 0.053 d |
| Chao1 | CK | 673.214 ± 21.296 ab | 543.470 ± 18.721 bc | 462.809 ± 3.810 b | 455.377 ± 60.937 a |
| | Gl | 773.434 ± 39.275 ab | 579.153 ± 19.726 abc | 464.205 ± 2.574 b | 479.181 ± 56.792 a |
| | Gm | 772.912 ± 28.255 a | 626.580 ± 29.634 a | 465.700 ± 4.486 b | 469.934 ± 53.990 a |
| | Gh | 717.335 ± 44.500 ab | 510.983 ± 12.810 c | 452.857 ± 1.069 b | 438.914 ± 55.892 a |
| | Dl | 733.530 ± 38.877 ab | 546.209 ± 15.359 bc | 461.212 ± 6.469 b | 438.962 ± 53.068 a |
| | Dm | 706.838 ± 26.245 ab | 595.042 ± 19.484 ab | 477.764 ± 9.637 b | 425.494 ± 44.871 a |
| | Dh | 641.645 ± 17.447 b | 578.217 ± 19.415 ab | 468.624 ± 8.868 b | 437.094 ± 48.999 a |
| | GDl | 697.382 ± 28.165 ab | 540.641 ± 15.796 bc | 452.803 ± 0.858 b | 455.715 ± 39.099 a |
| | GDm | 666.482 ± 30.530 b | 500.521 ± 2.086 bc | 496.728 ± 3.212 a | 491.149 ± 35.763 a |
| | GDh | 679.040 ± 27.961 ab | 541.952 ± 8.107 abc | 499.915 ± 16.673 b | 426.687 ± 6.304 a |

### 3.2.3. Beta Diversity of the Fungal Community

The principal coordinate analysis based on the Bray–Curtis distance for the fungal communities at the genus level is shown in Figure 6a–d. It can be observed that the differences in fungal community structure among different treatments increase over time ($R^2 \geq 0.847$, $p = 0.001$). It is worth noting that under low and high concentrations, the difference in community structure between the control treatment and the mixed herbicide treatment is smaller than the difference between the single herbicide treatment and the control ($R^2 \geq 0.534$, $p = 0.001$). From the above analysis, it can be concluded that the impact of mixed herbicides on the structure of soil fungal communities is less than that of single herbicides.

### 3.2.4. LEfSe Analysis of the Fungal Community

Linear discriminant analysis (LDA $\geq 2$) of the fungi is shown in Figures 4b and S3. In particular, at the phylum level, Ascomycota was extremely sensitive to the response of each herbicide treatment on day 28. It is worth noting that at the genus level, *Thielavia* had a sensitive response to GP on day 28, which was a significantly different species between GP treatment and other herbicides treatments. It can be concluded that GP significantly promoted the abundance of *Thielavia*, while there was no definite significant difference in species from mixed herbicide treatments when compared with the single herbicide treatments.

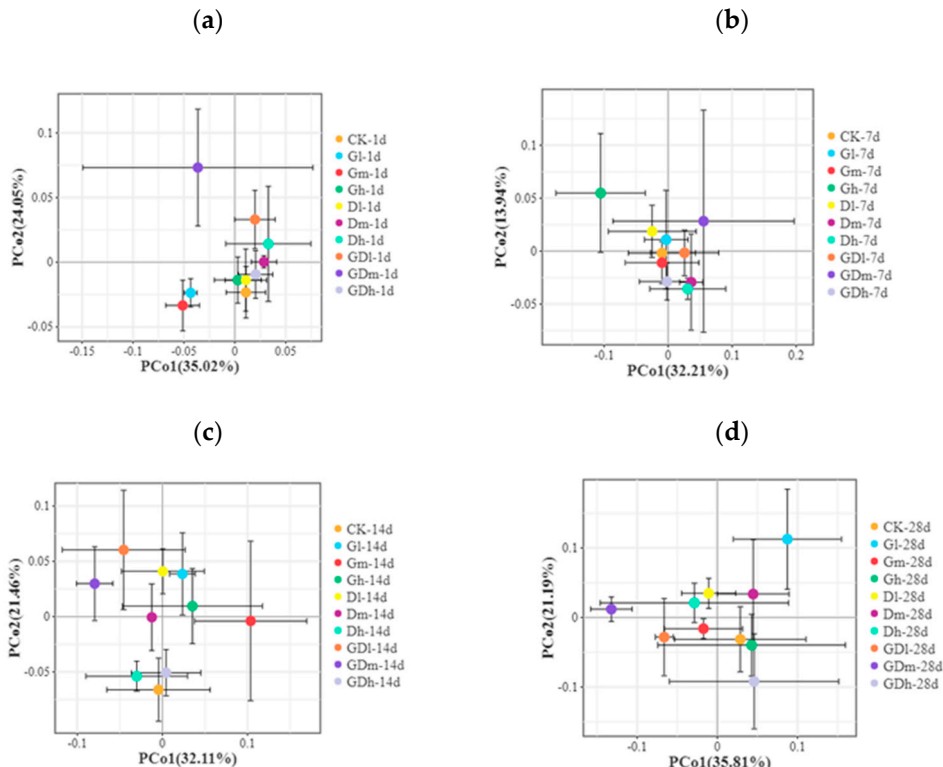

**Figure 6.** Principal coordinate analysis (**a**–**d**) for fungus communities at the genus level on 1-, 7-, 14- and 28-day exposure.

## 4. Discussion

Pesticide use has a harmful impact on soil biological activity, such as microbial abundance, diversity, and activities, all of which influence nutrient transformation and therefore, the health and quality of the soil [61]. This study found that the effect of combined pollution of GP and DQ on bacterial community structure at low concentrations was less than that of single herbicides. However, that was no difference at medium and high concentrations between mixed herbicides and single herbicides. In addition, the effect of mixed herbicides on the fungal community structure was less than that of single herbicide treatment. Most studies showed that herbicides have little or no short-term effect on the microbial community structure [29–31]. Previous studies have revealed that the continuous application of GP allows soil microorganisms to adapt to GP and that GP can select microbial populations capable of using it as a nutrient source, and the microbial community in the soil with long-term application of GP revealed a higher diversity index than that without the application of GP [62]. Studies on corn and soybean roots showed no effect of GP on the relative abundance of microbial organisms [63]. There is also literature that the microbial communities were negatively affected by GP [64] and the activity of total microbial community was also affected by GP [65]. In contrast, the ester-linked fatty acid methyl ester extraction (EL-FAME) analysis of agricultural soil exposed to repetitive application of GP displayed no significant changes in the structure of the soil microbial community [66]. In this study, the short-term effect of combined pollution of GP and DQ on microbial community structure was less than that of a single herbicide. Further research will be needed on the long-term effects of combined pollution of GP and DQ on soil microbial community structure.

The present study found that, compared with a single herbicide, the combined pollution of GP and DQ has a certain promoting effect on Actinobacteria and a certain inhibiting effect on Proteobacteria at the phylum level. Microbiological tests and cell metabolic response studies by Mara Grube et al. [66] showed that molasses can be used as a substrate to promote the growth of Actinobacteria in the presence of elevated concentrations of Gly; Actinobacteria was therefore considered resistant to elevated concentrations of Gly in

the growth environment and exhibited the potential for Gly degradation [62]. This may account for the increased abundance of Actinobacteria. It is well known that Actinobacteria and Proteobacteria are common bacteria taxa in soil [67], which may be sensitive to herbicide contamination; these taxa can have a variety of effects on soil and vegetation health, including beneficial and pathogenic effects [68,69]. At the genus level, the single herbicide decreased the relative abundance of *Streptomyces*. However, the combined pollution of GP and DQ has a certain promoting effect on *Streptomyces*. It was found that the combination of GP and Cu could reduce the toxicity of heavy metals to photoluminescent bacteria [46]; it may be that DQ and Cu combine with GP in a similar way to reduce the stress of some bacteria. Although GP promoted the abundance of *Sphingomonas*, the addition of GP enhanced the inhibitory effect of DQ on the abundance of *Sphingomonas*. In addition, single herbicides promoted or had no effect on the abundance of *Phenylobacterium*, while mixed herbicides had an inhibitory effect on the abundance of *Phenylobacterium*. Some researchers have suggested that the combined use of GP and Cd may aggravate the effects on *E. coli* [44]; perhaps, *Phenylobacterium* is as sensitive as E. coli to the stress of GP combined contamination. For fungi, although GP inhibited the abundance of *Talaromyces* at low and medium concentrations, the addition of GP enhanced the promotion of DQ on the abundance of *Talaromyces*. The single herbicide inhibited the abundance of *Curvularia*, while the mixed herbicides promoted the abundance of *Curvularia*. In addition, the abundance of *Conlarium* was significantly inhibited by mixed herbicides compared with single herbicides. Some studies have shown that fungal diversity and abundance respond strongly to high concentrations of GP, and herbicide combined pollution may make the stress response of different fungi more obvious.

The effects of combined pollution of GP and DQ on bacterial community richness and diversity were not significantly different from those of single herbicides. It was found that a single herbicide had a transient promoting or inhibiting effect on bacterial population abundance and community diversity in soil. Previous studies had also found that GP has an adverse effect on the interactions of manganese redox bacteria, *Pseudomonas fluorescens*, acetogenic rhizosphere bacteria and *Fusarium* in the rhizosphere soil of soybean, resulting in an increase in the number of *Fusarium* species, while the abundance of *Pseudomonas fluorescens*, manganese redox bacteria and acetogenic rhizosphere bacteria decreased [38]. However, the combined application of the two herbicides did not affect the richness and diversity of soil bacteria. This may be because some bacteria produce free radical scavenging molecules when they coexist with GP, and eliminate the free radicals produced by DQ, which makes the stress response of bacteria to GP and DQ reduced or unchanged [70]. Compared with single herbicides, mixed herbicides had no significant effect on the richness of soil fungal communities, but could inhibit the diversity of the fungal community. Some scholars had shown that GP can stimulate the soil fungal biomass in the early and short-term, and it has an adverse effect on both fungal community diversity and species richness after long-term application of GP [71]. In fact, the impact of GP on soil microbial communities and microbiota are highly variable and dependent upon specific experimental parameters such as the dose of GP applied, the time of incubation, and soil characteristics. In addition, the soil pH appears to regulate the balance between GP-induced toxicity and GP-induced microbial growth, with a lower pH favoring stimulation over suppression. In addition, there is a sensitivity spectrum in microbial population, such that less resilient species are inhibited by increasing GP concentrations, whilst a resistant degrader population compensates at higher concentrations [26]. Thus, mixed herbicides aggravated the adverse effects on soil fungal community diversity, but owing to the short period of this study, the effects of various herbicides on fungal population abundance were not significant.

Along with the persistence, concentration, toxicity, and bioavailability of the sprayed pesticide, a variety of environmental factors influence the toxic impact of pesticides on microbial diversity [72,73]. Here, we focused on the relative abundances and diversity of soil microbial community diversity only considering the concentration of herbicides.

Future studies should focus on the effects of combined pollution of GP and DQ in different soil environments considering the microbial community composition and diversity

## 5. Conclusions

This study examined the effects of the combined exposure of GP and DQ on the structure and diversity of microbial communities in lateritic paddy soil at the relative field application doses. Actinobacteria and Proteobacteria were the most sensitive microbial phyla with the application of mixed herbicides, which increased the abundance of Actinobacteria but significantly inhibited that of Proteobacteria, especially at low and medium concentrations. Compared with single herbicides, the mixed herbicide (GP + DQ) had no significant impacts on the richness and diversity of bacterial and fungal communities in the lateritic paddy soil. In general, the combined application of GP and DQ had no more adverse effects on soil microorganisms. Therefore, these two herbicides can be used reasonably in actual agricultural production.

**Supplementary Materials:** The following supporting information can be downloaded at: https://www.mdpi.com/article/10.3390/su15118497/s1, Figure S1: Principal coordinate analysis for the bacterial communities at the phylum level; Figure S2: Cladograms of line discriminant analysis effect size (LEfSe) analyses of bacteria; Figure S3: Cladograms of line discriminant analysis effect size (LEfSe) analyses of fungus; Table S1: Primer information. References [74–81] are cited in the supplementary materials.

**Author Contributions:** Conceptualization, methodology, writing—original draft preparation, X.H.; investigation, funding acquisition, C.W.; writing—review and editing, funding acquisition, H.T.; writing—review and editing, funding acquisition, X.D.; resources, Y.L. All authors have read and agreed to the published version of the manuscript.

**Funding:** This research was funded by the National Natural Science Foundation of China, grant number "42177402", the Natural Science Foundation of Hainan Province, grant number "420QN316" and the Hainan Province Science and Technology Special Fund of China "ZDYF2021XDNY137".

**Informed Consent Statement:** Not applicable.

**Data Availability Statement:** Not applicable.

**Conflicts of Interest:** The authors declare no conflict of interest.

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
