# Peer review of "Impact of Combined Exposure to Glyphosate and Diquat on Microbial Community Structure and Diversity in Lateritic Paddy Soil"

_sustainability, doi:10.3390/su15118497_

Round 1

Reviewer 1 Report

Please overview abstract and conclusion. There are some confusions regarding your results. 

Author Response

Please overview abstract and conclusion. There are some confusions regarding your results. 

Response: Thank you for your questions. We have carefully overview abstract and conclusion, and the confusions regarding the results have been modified in Abstract and Conclusion in the revised manuscript (Lines 21  45, Page 1; Lines 469  478, Page 14).

Abstract: Microbial communities play crucial role in the biogeochemical cycling of many important soil elements. Pesticides are known to affect non-targeted soil microorganisms. Glyphosate (GP) and diquat (DQ), as two commonly-used non-selective herbicides, are easily co-resided in lateritic paddy soil rich in iron/aluminum oxides. However, there is limited information on their co-impact on microbial community structure and diversity in this type of soil. Here, the short-term effects of combined exposure to GP and DQ on microbial diversity and community structure shifts were investigated in lateritic paddy soil from a tropical agricultural region (Hainan, China). The combined utilization dosages of two herbicides were set in three concentrations of low concentration (1-fold recommended dosage), medium concentration (10-fold recommended dosage) and high concentration (100-fold recommended dosage), respectively. The structure and diversity of microbial communities were determined via 16S rRNA and ITS gene high-throughput sequencing. The results revealed that Actinobacteria and Proteobacteria were the most sensitive microbial phyla to combined exposure to GP and DQ in lateritic paddy soil. Combined exposure to GP and DQ increased the abundance of Actinobacteria but significantly inhibit that of Proteobacteria, especially at low and medium concentrations. Compared with CK, mixed herbicide (GP + DQ) had no adverse effects on the richness of bacteria and fungi communities. and had no also adverse effects on the diversity of bacteria communities, but significantly decreased the diversity of fungi communities at high concentration within 28 days. However, The effects of combined exposure to GP and DQ on soil microbial richness and diversity were not significantly different from those of separate exposure of the two herbicides. In conclusion, the combined application of GP and DQ had no more adverse effects on soil microorganisms. Therefore, these two herbicides can be used reasonably in actual agricultural production.

Conclusions: This study examined the effects of combined exposure of GP and DQ on structure and diversity of microbial communities in lateritic paddy soil at the relative field application doses. Actinobacteria and Proteobacteria were the most sensitive microbial phyla to application of mixed herbicide. Which increased the abundance of Actinobacteria but significanltly inhibited that of Proteobacteria, especially at low and medium concentrations. Compared with single herbicide, the mixed herbicide (GP+DQ) had no significant impacts on the richness and diversity of bacteria and fungal communities in the lateritic paddy soil. In general, the combined application of GP and DQ had no more adverse effects on soil microorganisms. Therefore, these two herbicides can be used reasonably in actual agricultural production.

Line 28 - Commented [AI1]: Here you mentioned promoting effects of combined application, but in the conclusion, you mentioned no effect.

Response: The conclusions are supplemented with relevant results (Lines 472  473, Page 14).

Line 33 - Commented [AI2]: In the previous sentence, abundance of bacteria and fungi is mentioned in combined herbicidal treatment

Response: Thanks for the question, a more appropriate description has been revised into the manuscript (Lines 37  42, Page 1).

Line 43 - Commented [AI3]: Try to incorporate data for 2021-22

Response: The latest information was found in 2018 (Lines 50, Page 1).

Line 46 - Commented [AI4]: Need latest reference if possible

Response: We have tried our best to find the information.

Line 99 - Commented [AI5]: Is 10 is in the name of product?

Response: Yes. Nevertheless, the description of the product name is modified in the manuscript (Lines 114  115, Page 3).

Line 425-436 - Commented [AI6]: Not corelating with abstract. Please be clear either combined effect promoted microbial population or not

Response: The conclusion has corresponded to the abstract (Lines 469  478, Page 14). Results showing significant effects of mixed herbicides have been presented, but the regularity is not particularly strong, they can only be summarized generically.

Reviewer 2 Report

Well presented and well written manuscript. 

The abstract requires minor revision. Emphasis on the main key point(s) achieved from the combined use of herbicides. 

Again, be clear it is a comparative or combined-use study or both. 

In some parts of the results and discussion, it is unclear from the reviewer's perspective. 

Other part are well presented. 

Good work. Thanks.

Minor revision or perhaps proofreading suggested.  

Author Response

Well presented and well written manuscript. The abstract requires minor revision. Emphasis on the main key point(s) achieved from the combined use of herbicides. Again, be clear it is a comparative or combined-use study or both. 

In some parts of the results and discussion, it is unclear from the reviewer's perspective. 

Other part are well presented. 

Good work. Thanks.

Minor revision or perhaps proofreading suggested.

Response: Thank you for your valuable suggestions. We have emphasized the main key points achieved from the combined use of herbicides, and highlighted that this is a combination of comparative and combined-use study. Some of the descriptions of the results and discussions have been modified (Lines 450  457, Page 14; Lines 469  478, Page 14).

Abstract: Microbial communities play crucial role in the biogeochemical cycling of many important soil elements. Pesticides are known to affect non-targeted soil microorganisms. Glyphosate (GP) and diquat (DQ), as two commonly-used non-selective herbicides, are easily co-resided in lateritic paddy soil rich in iron/aluminum oxides. However, there is limited information on their co-impact on microbial community structure and diversity in this type of soil. Here, the short-term effects of combined exposure to GP and DQ on microbial diversity and community structure shifts were investigated in lateritic paddy soil from a tropical agricultural region (Hainan, China). The combined utilization dosages of two herbicides were set in three concentrations of low concentration (1-fold recommended dosage), medium concentration (10-fold recommended dosage) and high concentration (100-fold recommended dosage), respectively. The structure and diversity of microbial communities were determined via 16S rRNA and ITS gene high-throughput sequencing. The results revealed that Actinobacteria and Proteobacteria were the most sensitive microbial phyla to combined exposure to GP and DQ in lateritic paddy soil. Combined exposure to GP and DQ increased the abundance of Actinobacteria but significantly inhibit that of Proteobacteria, especially at low and medium concentrations. Compared with CK, mixed herbicide (GP + DQ) had no adverse effects on the richness of bacteria and fungi communities. and had no also adverse effects on the diversity of bacteria communities, but significantly decreased the diversity of fungi communities at high concentration within 28 days. However, The effects of combined exposure to GP and DQ on soil microbial richness and diversity were not significantly different from those of separate exposure of the two herbicides. In conclusion, the combined application of GP and DQ had no more adverse effects on soil microorganisms. Therefore, these two herbicides can be used reasonably in actual agricultural production.

The research have taken a different route by studying the combined used of two most well known herbicides. This itself is good and can be considered as original idea. SOme may have done so, however not well stated.

It does address a gap in the research field. The two herbicides are banned in some region and still used in some other regions. Thus it would be good for the scientific community to read the effect of such combined use application especially under paddy cultivation and lateritic soil. Because both this factors plays a vital role in the efficacy of the combined use.

Response: Thank you for your excellent comments.

A ew study idea with a twist on the combined used and community impact. That itself can be considered new if not fully novel or 100% novel. At least there are at least 80% novelty in the study.

Tables and figures are acceptable, and if any changes could be made, is the size and the colors of the table. If possible combined few figurea to show the combined use effect(s).

Response: Thank you for your suggestion. We have made some minor adjustments to the size of the table.

Reviewer 3 Report

1. Generally, the Introduction is good, but I would suggest to add more information regarding DQ effect on soil microbial community and groups. There is a lack of this information comparing to those about glyphosate. Also, it would be nice to add more references regarding the GP binding to soils (see notes below)

2. What were the hypothesis and the aim of the study?

3. Methods - please clarify how the sampled soils were stored before the analysis (temperature, time of storage).

Also, please describe how much soil was incubated with herbicides (the amount in grams in the microcosm or so), and how the samples were taken (samples for all days from 1 experimental microcosm? or were there any replications?)

4. Bioinformatic - please clarify, how the species annotation was made, and how diversity indices were calculated (programs, used databases, etc)

5. the description of figures are unclear - it should be like: CK - control, Gl - low concentration glyphosate, etc.

6. any suggestions why Actinobacteria was influenced by herbicide addition? some of the representatives could be potential degraders of glyphosate. Please check:

Grube, M., Kalnenieks, U., Muter, O., 2019. Metabolic response of bacteria to elevated concentrations of glyphosate-based herbicide. Ecotoxicol. Environ. Saf. 173, 373–380. doi:10.1016/j.ecoenv.2019.02.045

7. "It is worth noting" is overused throughout the manuscript

Other notes

Lines 24-25 - it would be good to add the herbicides doses here

Lines 27-34 - I recommend to split the sentence, because it sounds confusing

Line 49 - I suggest adding the information regarding the contamination of environments by DQ

Lines 49-51 - glyphosate also have a strong binding to Al and Fe oxides, please check the following references:

- Gimsing, A.L., Borggaard, O.K., Bang, M., 2004. Influence of soil composition on adsorption of glyphosate and phosphate by contrasting Danish surface soils. Eur. J. Soil Sci. 55, 183–191. doi:10.1046/j.1365-2389.2003.00585.x

- Mamy, L., Barriuso, E., 2005. Glyphosate adsorption in soils compared to herbicides replaced with the introduction of glyphosate resistant crops. Chemosphere 61, 844–855. doi:10.1016/j.chemosphere.2005.04.051

 (glyphosate adsorption in soils increased with ... iron and aluminium oxides content)

- Yu, Y., Zhou, Q.X., 2005. Adsorption characteristics of pesticides methamidophos and glyphosate by two soils. Chemosphere 58, 811–816. doi:10.1016/j.chemosphere.2004.08.064

Line 56 - small typing error in biochemistry (b, not B)

Line 58 - here specific microbial communities are mentioned; however, there should be specific functional or ecological microbial groups (because bacteria belonging to Acidobacteria or mycorrhizal fungi are groups, but not communities)

Lines 66-67 - this reference is relevant to aquatic environment

Line 154 - m.b. combined addition? in case of low doses it shoundn't be called pollution

Line 246 - please check the misspellings in the name of this part

Line 252 - should be "Chloroflexi"

Lines 363-365 - there are more recent studies regarding the glyphosate effect on microbial communities. Please check these references:

- Chavez-Ortiz, P., Tapia-Torres, Y., Larsen, J., Garcia-Oliva, F., 2022. Glyphosate-based herbicides alter soil carbon and phosphorus dynamics and microbial activity. Appl. Soil Ecol. 169. doi:10.1016/j.apsoil.2021.104256

- Kepler, R.M., Epp Schmidt, D.J., Yarwood, S.A., Cavigelli, M.A., Reddy, K.N., Duke, S.O., Bradley, C.A., Williams, M.M., Maula, J.E., 2020. Soil microbial communities in diverse agroecosystems exposed to the herbicide glyphosate. Appl. Environ. Microbiol. 86, 1–16. doi:10.1128/AEM.01744-19

Lines 411-413 - more useful information to this part of discussion could be found here:

Nguyen, D.B., Rose, M.T., Rose, T.J., Morris, S.G., van Zwieten, L., 2016. Impact of glyphosate on soil microbial biomass and respiration: A meta-analysis. Soil Biol. Biochem. 92, 50–57. doi:10.1016/j.soilbio.2015.09.014

Lines 421-423 - I suggest that future work should be focused more on verification of the data obtained by NGS using other methods, but that's only an opinion

Line 435 - no more adverse effect comparing to what? please clarify

Minor English editing is recommended

Author Response

1. Generally, the Introduction is good, but I would suggest to add more informationregarding DQ effect on soil microbial community and groups. There is a lack of this information comparing to those about glyphosate. Also, it would be nice to add more references regarding the GP binding to soils (see notes below)

Response: Thank you for your constructive suggestions. We did our best to obtain information regarding DQ effect on soil microbial community and groups, but useful information is very few. More references regarding the GP blinding to soils were added to the manuscript (Lines 60  61, Page 2).

2. What were the hypothesis and the aim of the study?

Response: Thank you for your questions. The hypothesis of this study is that mixed herbicides would have significant adverse effects on the overall community structure and diversity of soil microorganisms, in addition, the effects of mixed herbicides on soil microbiota were significantly different from those of single herbicides, and mixed herbicides can reduce or increase the abundance of certain microbial communities in the soil. The aim of the study is to explore whether the combined use of herbicides commonly used in tropical agricultural areas could lead to significant changes in the microflora of lateritic paddy soil in the region, to better manage and utilize herbicides commonly used in tropical agricultural areas. Related information was refined in the introduction (Lines 93  99, Page 2).

3. Methods - please clarify how the sampled soils were stored before the analysis (temperature, time of storage).

Also, please describe how much soil was incubated with herbicides (the amount in grams in the microcosm or so), and how the samples were taken (samples for all days from 1 experimental microcosm? or were there any replications?)

Response: Thank you for your questions. The sampled soils were stored at -80 °C and taken out for analysis within 3 days. The initial weight of soil before incubating with herbicides in each treatment was 50 g. The sample collection is based on the following literature:

Wu, C., Wang, Z., Ma, Y., Luo, J., Gao, X., Ning, J., Mei X., She, D. (2020). Influence of the neonicotinoid insecticide thiamethoxam on soil bacterial community composition and metabolic function. Journal of Hazardous Materials, 124275. doi:10.1016/j.jhazmat.2020.124275

With regard to the above-mentioned issues, the material part of the manuscript has been revised (Lines 122  126, Page 3).

4. Bioinformatic - please clarify, how the species annotation was made, and how diversity indices were calculated (programs, used databases, etc)

Response: Thank you for your valuable suggestions. The representative OTU sequences or ASV sequences were classified into organisms by a naive Bayesian model using RDP classifier [1] (version 2.2) based on SILVAdatabase [2] (version 132) or UNITEdatabase [3] (version 8.0) or ITS2 database [4] (version update_2015), with the confidence threshold value of 0.8.

Chao1, Shannon’s evenness index were calculated in QIIME [5] (version 1.9.1). Alpha index comparison between groups was calculated by Welch's t-test andWilcoxonrank test in R project Vegan package [6] (version 2.5.3). Alpha index comparison among groups was computed by Tukey’s HSD test andKruskal-WallisH test in R project Vegan package [6] (version 2.5.3).  

With regard to the above we have made a detailed supplement of the material section of the manuscript (Lines 157  167, Page 4).

[1] Wang, Qiong, et al. Naive Bayesian classifier for rapid assignment of rRNAsequences into the new bacterial taxonomy. Applied and environmental microbiology 73.16 (2007): 5261-5267.

[2] Pruesse, Elmar, et al. SILVA: a comprehensive online resource for quality checked andaligned ribosomal RNA sequence data compatible with ARB. Nucleic acids research 35.21 (2007): 7188-7196.

[3] Nilsson R H, Larsson K H, TaylorAF S, et al. The UNITE database for molecular identification of fungi: handling dark taxa and parallel taxonomic classifications. Nucleic acids research, 2018, 47(D1): D259-D264.

[4] Ankenbrand M J, Keller A, Wolf M, et al. ITS2 database V: Twice as much. Molecular Biology and Evolution, 2015, 32(11): 3030-3032.

[5] Caporaso, J. Gregory, et al. QIIME allows analysis of high-throughput community sequencing

data. Nature methods 7.5 (2010): 335-336.

[6] Oksanen J, Blanchet F G, Kindt R, et al. Vegan: community ecology package. R package

version 1.17-4. http://cran. r-project. org>. Acesso em, 2010, 23: 2010.

5. The description of figures are unclear - it should be like: CK - control, Gl - low concentration glyphosate, etc.

 Response: The description of figures have been modified in the revised manuscript (Lines 203  207, Page 5).

6. Any suggestions why Actinobacteria was influenced by herbicide addition? some of the representatives could be potential degraders of glyphosate. Please check:

Grube, M., Kalnenieks, U., Muter, O., 2019. Metabolic response of bacteria to elevated concentrations of glyphosate-based herbicide. Ecotoxicol. Environ. Saf. 173, 373–380. doi:10.1016/j.ecoenv.2019.02.045

Response: Thank you for your constructive suggestions. Acoording to this reference, the results in our study might attributed to Actinobacteria that is resistant to increasing amounts of glyphosate herbicides in its growing environment and exhibit degradation potential for glyphosate, and the information has been provided in the revised manuscript (Lines 405 - 410, Page 13).

7. "It is worth noting" is overused throughout the manuscript

Other notes

Response: Good suggestion. The irrational phrase in the manuscript have been checked and corrected.

Lines 24-25 - it would be good to add the herbicides doses here

Response: The herbicides doses have been added (Lines 28  31, Page 1).

Lines 27-34 - I recommend to split the sentence, because it sounds confusing

Response: Corrected (Lines 33  43, Page 1).

Line 49 - I suggest adding the information regarding the contamination of environments by DQ

Response: Related information has been added, but we are so sorry that we did not get more available information about DQ environmental contamination (Lines 56  58, Page 2).

Lines 49-51 - glyphosate also have a strong binding to Al and Fe oxides, please check the following references:

- Gimsing, A.L., Borggaard, O.K., Bang, M., 2004. Influence of soil composition on adsorption of glyphosate and phosphate by contrasting Danish surface soils. Eur. J. Soil Sci. 55, 183–191. doi:10.1046/j.1365-2389.2003.00585.x

- Mamy, L., Barriuso, E., 2005. Glyphosate adsorption in soils compared to herbicides replaced with the introduction of glyphosate resistant crops. Chemosphere 61, 844–855. doi:10.1016/j.chemosphere.2005.04.051

 (glyphosate adsorption in soils increased with ... iron and aluminium oxides content)

- Yu, Y., Zhou, Q.X., 2005. Adsorption characteristics of pesticides methamidophos and glyphosate by two soils. Chemosphere 58, 811–816. doi:10.1016/j.chemosphere.2004.08.064

Response: Thank you for your advice. We consulted the literature you provided (Lines 60  61, Page 2).

Line 56 - small typing error in biochemistry (b, not B)

Response: Corrected (Lines 68, Page 2).

Line 58 - here specific microbial communities are mentioned; however, there should be specific functional or ecological microbial groups (because bacteria belonging to Acidobacteria or mycorrhizal fungi are groups, but not communities)

Response: Thank you for your correction. The description of the relevant information has been modified (Lines 77  78, Page 2).

Lines 66-67 - this reference is relevant to aquatic environment

Response: This reference has been deleted.

Line 154 - m.b. combined addition? in case of low doses it shoundn't be called pollution

Response: Agree. Corrected (Lines 178, Page 4; Lines 294, Page 9).

Line 246 - please check the misspellings in the name of this part

Response: Thank you for your correction. It has been corrected in the manuscript (Lines 271, Page 8).

Line 252 - should be "Chloroflexi"

Response: Corrected (Lines 277, Page 8).

Lines 363-365 - there are more recent studies regarding the glyphosate effect on microbial communities. Please check these references:

- Chavez-Ortiz, P., Tapia-Torres, Y., Larsen, J., Garcia-Oliva, F., 2022. Glyphosate-based herbicides alter soil carbon and phosphorus dynamics and microbial activity. Appl. Soil Ecol. 169. doi:10.1016/j.apsoil.2021.104256

- Kepler, R.M., Epp Schmidt, D.J., Yarwood, S.A., Cavigelli, M.A., Reddy, K.N., Duke, S.O., Bradley, C.A., Williams, M.M., Maula, J.E., 2020. Soil microbial communities in diverse agroecosystems exposed to the herbicide glyphosate. Appl. Environ. Microbiol. 86, 1–16. doi:10.1128/AEM.01744-19

Response: Thank you for your advice. These references have been cited in the manuscript (Lines 390  395, Page 12).

Lines 411-413 - more useful information to this part of discussion could be found here:

Nguyen, D.B., Rose, M.T., Rose, T.J., Morris, S.G., van Zwieten, L., 2016. Impact of glyphosate on soil microbial biomass and respiration: A meta-analysis. Soil Biol. Biochem. 92, 50–57. doi:10.1016/j.soilbio.2015.09.014

Response: Thank you for your valuable advice. The reference has been cited in the manuscript (Lines 405  410, Page 13).

Lines 421-423 - I suggest that future work should be focused more on verification of the data obtained by NGS using other methods, but that's only an opinion

Response: Thank you for your valuable suggestions. The opinion will be take into consideration in our future work.

Line 435 - no more adverse effect comparing to what? please clarify

Response: Thank you for your question. The combination of GP and DQ had no more adverse effect on soil microorganisms compared to single herbicides. The relevant information has been clarified in the manuscript (Lines 474  476, Page 14).

Minor English editing is recommended

Response: Thank you for your advice. Native English speakers helped to examine the manuscript.

Reviewer 4 Report

This manuscript investigated the short-term combined effects of glyphosate and diquat on soil microbial activities and found that glyphosate and diquat had inhibitory effects on the abundance of Streptomyces and Curvularia. This manuscript was well written. I recommended to accept it after minor revision. 

The quality of English language was in the journal standard

Author Response

This manuscript investigated the short-term combined effects of glyphosate and diquat on soil microbial activities and found that glyphosate and diquat had inhibitory effects on the abundance of Streptomyces and Curvularia. This manuscript was well written. I recommended to accept it after minor revision. 

The quality of English language was in the journal standard

Response: Thank you for your affirmation and your constructive suggestions for this paper. The manuscript has been revised to make the results seem more reasonable.